# A Review on Locomotion Mode Recognition and Prediction When Using Active Orthoses and Exoskeletons

**DOI:** 10.3390/s22197109

**Published:** 2022-09-20

**Authors:** Luís Moreira, Joana Figueiredo, João Cerqueira, Cristina P. Santos

**Affiliations:** 1Center for Microelectromechanical Systems (CMEMS), University of Minho, 4800-058 Guimarães, Portugal; 2LABBELS—Associate Laboratory, 4800-058 Guimarães, Portugal; 3Life and Health Sciences Research Institute (ICVS), University of Minho, 4800-058 Guimarães, Portugal; 4Clinical Academic Center (2CA-Braga), Hospital of Braga, 4700-099 Braga, Portugal

**Keywords:** gait rehabilitation, locomotion mode recognition and prediction, wearable assistive devices

## Abstract

Understanding how to seamlessly adapt the assistance of lower-limb wearable assistive devices (active orthosis (AOs) and exoskeletons) to human locomotion modes (LMs) is challenging. Several algorithms and sensors have been explored to recognize and predict the users’ LMs. Nevertheless, it is not yet clear which are the most used and effective sensor and classifier configurations in AOs/exoskeletons and how these devices’ control is adapted according to the decoded LMs. To explore these aspects, we performed a systematic review by electronic search in *Scopus* and *Web of Science* databases, including published studies from 1 January 2010 to 31 August 2022. Sixteen studies were included and scored with 84.7 ± 8.7% quality. Decoding focused on level-ground walking along with ascent/descent stairs tasks performed by healthy subjects. Time-domain raw data from inertial measurement unit sensors were the most used data. Different classifiers were employed considering the LMs to decode (accuracy above 90% for all tasks). Five studies have adapted the assistance of AOs/exoskeletons attending to the decoded LM, in which only one study predicted the new LM before its occurrence. Future research is encouraged to develop decoding tools considering data from people with lower-limb impairments walking at self-selected speeds while performing daily LMs with AOs/exoskeletons.

## 1. Introduction

Humans can usually adjust their locomotion mode (LM) according to a variety of conditions and terrains that they typically face. LMs are composed of static and dynamic tasks. The static tasks correspond to the sitting (SIT) and standing (ST) tasks, while the dynamic tasks are further divided into two more categories: continuous and discrete. The continuous tasks correspond to the level-ground walking (LW), stair ascending (SA) and descending (SD), and ramp ascending (RA) and descending (RD) tasks. On the other hand, the discrete tasks consist of transitions between tasks. All these tasks can be recognized after their occurrence or predicted before it. However, patients with lower-limb impairments face challenges while performing these daily tasks [1].

Current challenges in personalized robotics-based assistance are related to recognizing and predicting different LMs with a non-intrusive sensor setup to timely trigger the assistance delivered by the wearable assistive devices. Despite recent advancements, most of the current LM decoding tools integrated into wearable assistive devices (i) address a limited number of daily LMs (are non-generic tools) [2,3]; (ii) present high recognition delays [4,5,6,7] and low prediction times [6,8,9,10]; (iii) do not consider the effect of the walking speed variation [6,8,11,12]; (iv) do not present clinical evidence [3,6,8,13]; and (v) do not adapt the assistance to perform the intended motion [3,6,7,8,13]. It is of the utmost importance that wearable assistive devices, such as AOs and exoskeletons, tackle these limitations by endowing algorithms capable of accurately and timely decoding of the user’s LMs to provide personalized assistance [7,14].

Reviews already published provide an overview of the state of the art of wearable assistive devices [1,14,15]. Yan et al. [15] focused on assistive control strategies for lower-limb AOs and exoskeletons, stating the available devices are able to assist users with lower-limb impairments and specifying their control strategies. Despite providing a valuable contribution to the existing assistive devices, topics related to (i) the control strategies oriented to the user’s needs and (ii) for which sensors and machine learning classifiers should be employed to decode LM were not deeply explored. On the other hand, Novak et al. [14] provide the strengths and weaknesses of using different sensors (electromyography (EMG), electroencephalography (EEG), and mechanical sensors) and their fusion in several applications, such as LM decoding. However, this review [14] focuses on which sensors can be applied to some applications, including AOs and exoskeletons, and it does not address how to use the output of the LM decoding tools to control these wearable assistive devices. Labarrière et al.’s [1] review is centered on the sensors and machine learning classifiers used to recognize and predict LMs when using AOs, exoskeletons, and prostheses. Nonetheless, most of the studies presented in [1] were designed for assisted conditions when using prostheses [16,17,18,19,20]. According to [11], the LM recognition performed when using orthotic/exoskeleton systems is different from prostheses or conditions without a lower-limb assistive device (non-assisted conditions [21,22,23,24,25,26,27]) due to the distinct positions that the sensors can take in these situations. Additionally, the human–robot interaction is also different between orthotic/exoskeleton systems and prostheses, mainly when the motion activity of the lower limb is limited [28].

This review addresses LM decoding tools developed and applied exclusively in orthotic systems and/or exoskeletons. Thus, this review advances the studies [1,14,15] by analyzing (i) the sensors and their positioning during LMs applications when using AOs and/or exoskeletons; (ii) the machine learning classifiers applied for LM decoding; and (iii) the control architecture adopted by the wearable assistive devices when recognizing and/or predicting LMs, providing an overview of the current ability of these devices to assist according to the users’ motion intentions. Four research questions (RQs) related to LM decoding under robotics assistance were identified in this literature review: (RQ1) Which are the typical LMs and the target population addressed?; (RQ2) Which type of wearable sensors and features are commonly used for LM recognition and prediction?; (RQ3) Which set of algorithms should be employed to recognize/predict different LMs attending to accuracy and time-effectiveness?; and (RQ4) how to adapt the exoskeleton/orthosis assistance according to the decoded user’s LM.

## 2. Methods

### 2.1. Search Strategy

The literature search was conducted from 1 December 2020 to 31 August 2022 in the *Scopus* and *Web of Science* databases using the following keywords: “locomotion mode recognition”; “locomotion mode prediction”; “locomotion mode transition”; “locomotion transition” or “locomotion prediction”; “motion intention decoding”; “motion intention” and “orthosis”; “locomotion mode recognition” and “orthosis”; “locomotion mode recognition” and “exoskeleton”; “motion intention recognition” and “orthosis”; and “motion intention recognition” and “exoskeleton”. Variations of “intention” to “intent” were also considered. This search was limited to titles, keywords, and abstracts.

### 2.2. Eligibility Criteria

Manuscripts were evaluated based on the following inclusion criteria: (i) the study was published in English; (ii) the study is after 2010; (iii) the study focuses on lower-limb LM decoding in real time; (iv) the LM decoding algorithm only relies on data acquired from wearable sensors; and (v) only wearable assistive devices, such as exoskeletons and AOs, should be included. All the studies where the participants were not wearing any wearable assistive device (exoskeletons or AOs) or using prosthetic systems were excluded. Reviews were also excluded from the analysis.

### 2.3. Data Extraction

The information extracted from each study focused on five main topics, namely: (i) the addressed motion tasks and walking speed; (ii) the sensors used, the features extracted, and the windows employed to extract features; (iii) the classifiers and their performance in terms of accuracy and decoding delay; (iv) the control type of the assistive device considering the decoded LM; and (v) the number and condition (healthy or pathology) of the participants involved for developing the decoding tool.

### 2.4. Quality Assessment

The quality of the included studies was evaluated based on a modified QualSyst Tool presented in [29]. Twelve criteria were employed, in which criteria 1 to 4, 7, 11, and 12 are the same as stated in [29]. Criteria 8, 9, and 10 of [29] correspond to criteria 6, 8, and 9 of this manuscript, respectively. Criteria 5 and 6 of [29] were removed and replaced by criteria 5 and 10 to include items essential to the LM recognition/prediction and correspondent assistance by wearable assistive devices, such as the sensors used, data collected, and control strategies developed. The authors approved the proposed criteria.

The twelve criteria are stated in Appendix A and detailed in Table A1. The first two criteria evaluated if the objective, research questions, and study design were adequately described and appropriate. Criterion 3 analyzed if the participants’ demographic characteristics and the inclusion and exclusion criteria were clearly stated. Criterion 4 evaluated if the experimental protocol of each study was presented with enough rigor to enable reproducibility, while criterion 5 analyzed if the sensors used and the data collected were clearly reported. Criteria 6 and 7 checked if the studies provided enough information about the input features and windows length, respectively. Criteria 8 and 9 confirmed that the classification algorithms and evaluation methods were clearly presented. Criterion 10 evaluated if the control strategy of the wearable assistive device used in each study was mentioned and explained. Criterion 11 analyzed if the study results were given, stating the mean and standard deviation values, and criterion 12 checked if the conclusions were drawn based on all results, including positive and negative results.

Each one of the included papers was evaluated with a score ranging from 0 to 2. A score of 0 implies an inexistence of information, whereas a score of 1 and 2 represent partial and full information, respectively. As the tool includes 12 criteria, the minimum and maximum scores that each study can achieve are 0 and 24, respectively. These scores were converted on a scale from 0 to 100%.

## 3. Results

### 3.1. Studies Selection

This literature search resulted in 119 and 136 studies from *Scopus* and *Web of Science* databases, respectively. Thirteen papers were manually identified in the references section of other studies and accounted for this analysis, leading to 268 identified papers. After removing the duplicate papers, 129 studies remained for screening, of which 79 papers were removed based on the titles and abstracts. A total of 60 full-text articles were assessed for eligibility, and according to the inclusion and exclusion criteria, 18 studies were included. Figure 1 depicts the PRISMA flow chart discretizing this selection process.

### 3.2. Quality of the Included Studies

Table 1 presents the quality assessment analysis of the included studies computed based on the modified QualSyst Tool. The mean, maximum, and minimum quality scores obtained were 84.7 ± 8.7%, 95.8%, and 62.5%, respectively.

### 3.3. Information Extracted from the Included Studies

Table 2 presents the information extracted from each study, attending to the addressed locomotion tasks, speed, sensors used, features extracted, and analysis windows employed. Information regarding the classifiers and their performance in terms of accuracy and decoding delay is also provided, along with the control type adopted by the wearable assistive device considering the decoded LM. The last column presents the number and status (healthy or pathological) of the participants involved in the algorithms’ development and validation.

#### 3.3.1. Locomotion Modes and Speed

Based on Table 2, the most investigated LMs are dynamic tasks only [2,4,5,8,10,13,30,31,32] (N = 11) and a combination between dynamic and static tasks [3,6,9,11,12,34,35] (N = 7). Among the dynamic tasks, six studies [4,13,31,33,34,35] classified continuous dynamic tasks, whereas only one study [2] focused on transitions. Thirteen studies [3,5,6,7,8,9,10,11,12,30,32,34,35] explored continuous and discrete dynamic tasks together.

The ST task was the most investigated static LM [3,6,7,9,11,12,34,35] (N = 8), whereas the SIT task was explored in three studies [6,7,35]. Three studies [6,7,35] considered the identification of both ST and SIT tasks.

Concerning the dynamic LMs, the LW, SA, and SD tasks were the continuous dynamic tasks typically addressed, being explored in 17 studies [3,4,5,6,7,8,9,10,11,12,13,30,31,32,33,34,35], while the RA and RD tasks were investigated together with the LW, SA, and SD tasks in nine studies [4,8,11,12,13,30,31,32,35]. The LW-SA, SA-LW, LW-SD, and SD-LW transitions were addressed in eight studies [5,6,7,8,9,10,30,32], being the most investigated discrete dynamic tasks. The LW-RA, RA-LW, LW-RD, and RD-LW tasks were explored in four studies [8,9,30,32], the SIT-ST and ST-SIT in four studies [2,6,7,35], the ST-LW and LW-ST in two studies [6,7], and the ST-SA and SA-ST in one study [6].

The LMs regarding the LW, SA, SD, RA, and RD tasks together and the transitions between them were decoded in three studies [8,30,32]. Moreover, only two studies [6,34] considered the walking initiation and termination motions. Finally, no study has explored the ST-RA, RA-ST, ST-RD, and RD-ST transitions.

Only four studies predicted tasks before their occurrence, namely (i) the ST task preceded by the SIT task and vice-versa [6]; (ii) the LW task preceded by the SA task and vice-versa [8,9]; (iv) the LW task preceded by the SD task [8,9,10]; and (v) the SD task preceded by the LW task [9]. However, no study was able to predict all the proposed LMs.

Ten studies [3,5,6,7,8,11,12,13,32,33] presented information regarding the walking speed during the LW task, whereas only four studies presented its value (1.0 [33], 1.5 [33], 2.7 [11,12], and 4.0 km/h [13]). While in [13,33], the presented speed value was employed during the LW tasks, in [11,12], the speed of 2.7 km/h was enforced during the LW, RA, and RD tasks, using a treadmill. In the remaining eight studies, the participants walked at self-selected speeds, but there was no information regarding the mean value of the self-selected speed. Study [7] asked participants to walk at self-selected slow, natural, and fast speeds.

#### 3.3.2. Sensor Systems, Features, and Analysis Windows

Based on the information presented in Table 2, (i) one study only used physiological sensors [33]; (ii) nine studies have only used kinematic sensors [2,3,5,9,10,11,12,30,34]; (iii) eight studies have combined kinematic and kinetic sensors [4,6,7,8,13,30,31,35]; (iv) no study used kinetic sensors exclusively; and (v) no studies combined these three sensors’ types.

A single study used an EMG system, acquiring signals from the vastus lateralis, vastus medialis, semitendinosus, and semimembranosus muscles to map the user’s motion intention [33].

Seventeen studies used kinematic sensors, namely (i) potentiometers [2,3] (N = 2) and encoders [6,7,13,30] (N = 4) embedded in the joints of the wearable assistive device to measure the joint angles and (ii) IMUs [3,4,5,9,10,11,12,13,30,31,32,34,35] (N = 13) and Attitude and Heading Reference System (AHRS) [8] (N = 1) placed on both wearable assistive device and participants’ lower limbs to measure the inclination angles, angular velocities, accelerations, and positions of the segments. In addition, eight studies combined kinematic with kinetic sensors, such as (i) encoders and pressure insoles [6,7] (N = 2); (ii) encoders, IMUs, and load cells [13,35] (N = 2); (iii) IMUs and Force Sensing Resistors (FSRs) [4,31] (N = 2); and (iv) encoders, IMUs, and specific Ground Reaction Force (GRF) sensors [30] (N = 1), which are commonly embedded in the wearable assistive device. These kinetic sensors measure the human–robot interaction, foot contact with the ground, the force resultant from that contact, and the center of pressure (CoP). Figure 2 represents all mentioned sensors considering their positioning in the leg with and without assistance.

In relation to the features that serve as input to the classification algorithms, two types were found: (i) time-domain data [2,3,4,5,6,7,9,10,11,12,13,30,31,32,33,34,35] (N = 17) and (ii) frequency-domain data [8] (N = 1). Of the 18 studies, 9 used an analysis window to compute the features [5,7,8,9,10,11,12,34,35]. In these studies [5,7,8,9,10,11,12,34,35], the extracted time-domain features were (i) the hip joint angles and CoP at the heel strike event [7]; (ii) maximum and minimum thigh, shank, and knee angles [5]; and (iii) maximum, minimum, mean, standard deviation, and root mean square of the thigh and shank inclination angles, triaxial angular velocities, and accelerations [9,10,11,12,34,35]. The time-frequency domain features considered were the wavelet coefficients from the (i) GRF during the swing phase [8] and (ii) thigh and foot inclination angles [8]. Seven studies defined a sliding window, characterized by a window length (from 100 to 250 ms) and an increment (from 10 to 50 ms) [5,8,10,11,12,34,35]. Furthermore, two studies only specified the window length, using 150 ms in [9] and 200 ms in [7]. All of the remaining studies [2,3,4,5,6,13,30,31] (N = 8) did not provide information about analysis windows. Instead, they used raw temporal information extracted from sensors: (i) hip, knee, and ankle joint angle depending on the assisted joint [2,3,6,32]; (ii) inclination angles of the thigh, shank, and foot [4,13,31]; and (iii) vertical position and velocity of the striking foot [3,13,31].

#### 3.3.3. Classifiers

Ten different algorithms were applied, namely: event-based fuzzy-logic methods, decision tree (DT), support vector machine (SVM), discriminant analysis (DA), k-nearest neighbor (kNN), ensemble method (EM), convolutional neural network (CNN), stacked autoencoder-based deep neural network, multilayer feedforward neural network (MFNN), and finite state machine (FSM). Two algorithms were employed more times than others: FSM [3,4,5,6,7,9,34] (N = 7) and SVM [2,8,9,10] (N = 4). The remaining classification algorithms were used three times (MFNN [11,12,31], event-based fuzzy-logic methods [6,7,34], and CNN [30,32,35]), twice (DT [13,30]), and once (DA [30], kNN [30], EM [30], and stacked autoencoder-based deep neural network [30]).

Sixteen studies used the accuracy and the recognition delay to assess the performance of the classifiers. From these 16 studies, all present accuracy values above 90%. In particular, 12 exhibited classification accuracies higher than 95% [3,4,5,7,8,9,11,12,13,30,31,32]. In general, concerning the recognition delay, the LMs were recognized between 3.96% and 100% of the gait cycle of the new LM.

There were four studies able to predict some tasks before their occurrence [6,8,9,10]. In [6], the ST task preceded by the SIT task (and vice-versa) was predicted using a FSM with a delay of −30.9% and −31.7%, respectively. No standard deviation was provided. The negative sign means that the decoding was done before the LM occurrence, i.e., in the gait cycle that precedes the one of the new LM. In [8], the SA task preceded by the LW tasks (and vice-versa) was predicted using the SVM classifier with a recognition delay of −10.4 ± 1.2% and −6.40 ± 0.8%, respectively. In [9], the SVM classifier predicted (i) the SA task preceded by LW task in −5.4 ± 74.6 ms; and (ii) the SD preceded by LW task in −78.5 ± 25.0 ms when the leading leg was the leg without the wearable assistive device. When the leading leg was the paretic leg (with the wearable assistive device), the study was able to predict the (i) SD task preceded by LW task in −1.8 ± 39.2 ms; (ii) LW task preceded by SD task in −0.7 ± 58.7 ms; and (iii) LW task preceded by SA task in −46.7 ± 46.6 ms. In [10], the SD task preceded by LW task was predicted 40.0 ± 107.53 ms before its occurrence, using the SVM classifier.

Figure 3a–d shows an overview of the relationship between the average accuracy obtained from decoding a specific LM and the used classifiers and sensors. Nonetheless, a direct comparison regarding the accuracy of each study cannot be made since the protocol and the classified LMs differ among the studies.

Table 2 shows that the MFNN [31], FSM [4], and DT [13] classifiers were employed to decode continuous dynamic tasks. The average accuracies are depicted in Figure 3a. Depending on the sensors used and the LMs to decode, these classifiers appear to provide a high capability of decoding continuous LMs. Moreover, the recognition delay for each study was (i) between 16% and 28% [31]; (ii) of one-step delay [4]; and (iii) below 5.13% [13], respectively (Table 2).

The MFNN [11,12], and FSM [3] were employed to decode static and continuous dynamic tasks together. Similar accuracies were reported for MFNN and FSM algorithms even using IMU sensors or combining IMUs with potentiometers (Figure 3b). Regarding the recognition delay, no information was provided in [11]. A one-step delay and a delay between 50 and 300 ms were reported in [3] and [12], respectively.

A greater variety of classifiers have been employed to decode continuous and discrete dynamic tasks together, namely SVM [8,9,10], DA [30], DT [30], kNN [30], EM [30], CNN [30,32], stacked autoencoder-based deep neural network [30], and FSM [5]. Based on the average accuracies presented in Figure 3c, SVM, kNN, CNN, stacked autoencoder-based deep neural network, and FSM seem to provide the highest performances. It is noteworthy that SVM [8,9,10] and CNN [30,32] algorithms reported different classification accuracies depending on the sensors employed. Moreover, these classifiers reported higher classification accuracies when only IMU data were used as input.

Two classifiers were used to decode static, continuous, and discrete dynamic tasks together, namely the FSM combined with the event-based fuzzy-logic method [6,7,34], the FSM combined with SVM [9], and CNN [35]. In general, similar average accuracies were reported for both algorithms, as presented in Figure 3d.

#### 3.3.4. Control Type of the Wearable Assistive Device

LM decoding tools were developed under the assistive device action managed by three control modes: (i) zero-torque mode [2,3,5,6,7,8,10,11,12,13,31,34,35] (N = 13); (ii) assistive mode without considering the response of the LM recognition/prediction tools [11] (N = 1) and (iii) assistive mode while considering the response of the LM recognition/prediction tools to support the users according to their intent [4,7,9,32,33,34] (N = 6).

The zero-torque mode implies that the wearable assistive device is operating passively, enabling it to be freely moved by the user. This mode is also known as the transparent mode, in which constraints associated with the actuator’s frictions are reduced [9]. The assistive mode without considering the LM recognition/prediction tools corresponded to the employment of torque controls, in which the wearable assistive device is enforcing a pattern to the user [12]. On the other hand, the assistive mode considering the LM recognition/prediction tools also enforces a pattern to the user but commonly takes advantage of a three or two-level control architecture to assist the user according to the LM decoded [41]. The three-level control scheme is characterized by a (i) high-level or perception layer, employing recognition algorithms to decode the user’s LMs; (ii) mid-level or conversion layer that maps the user’s motion intention to the control algorithm to generate reference trajectories according to the desired/intended motion; and (iii) the low-level or execution layer, in which the control algorithm drives the wearable assistive device to perform the desired motion [41]. In the two-level control scheme, there is a high and low level. While the low level is the same as in the three-level control scheme, the high level corresponds to the combination of the high and middle level of the three-level control scheme [7].

Six studies have adapted the assistance of the wearable assistive device attending to the user’s decoded LM [4,7,9,32,33,34]. In [4,7,34], a two-level hierarchical control design was followed. In [4], a constant torque of 10 N.m was provided by an ankle orthosis (low level) when the SD task was recognized (high level). For that, FSRs sensors embedded in the orthosis foot were used to detect the gait phase. Then, the LM was decoded, and the constant torque was provided. The torque was delivered during (i) the first 50% of the gait cycle to support the ankle joint motion when the leg without assistance is in the swing phase and (ii) the last 20% of the gait cycle to guarantee that the foot is performing plantarflexion before touching the next stair step. In [7], a two-level hierarchical control was designed to provide a phase-locked assistive torque (low-level) when a LM (among the SIT, ST, LW, SA, and SD tasks and transitions between them) was recognized (high level). Contrarily to [4], in [7], after recognizing the LW, SA, and SD tasks, a gait phase estimation algorithm was employed to provide the phase-locked assistive torque most suitable according to the gait phase of the decoded LM (Figure 4). The provided torque was adopted from public databases with joint trajectories recorded during locomotion-related activities [42,43,44,45]. Moreover, the torque patterns were modeled according to each participant’s body mass. Similarly to [7], a predefined joint torque curve was adopted according to the decoded LM in [34].

In [9,32,33], a three-level hierarchical control architecture was followed. In [32], the high level presented a gait phase estimation algorithm using IMU signals to segment the input data into strides. Based on these strides, the LM was decoded. A parameter optimal iterative learning control (POILC) method [46] was implemented in the middle level. This method was designed for a soft lower-limb exoskeleton to provide hip and knee assistance according to the tension of Bowden cables. The tension of these cables was modeled according to the hip and knee joint moments when performing the LW, SA, and SD tasks [47]. The low level was responsible for driving the exoskeleton according to the tension of the Bowden cables. The high level developed in [9] was different from [32]. Firstly, the LM was distinguished between a static (ST) and a dynamic condition. Then, in the case of a dynamic condition, the LM was classified (among LW, SA, and SD tasks), followed by a gait phase estimation algorithm to distinguish the stance from the swing phase. Three other subphases (heel-strike, heel-off, and toe-off) were identified during the stance phase. In addition, the mid-level developed in [9] was hybrid. It used a zero-torque mode during the ST task and during the swing phase of LW, SA, and SD tasks in an attempt to reduce the constraints associated with the actuator’s frictions. According to the authors [9], during the first and second subphases (from the heel-strike to heel-off events and from heel-off to toe-off events, respectively) of the SA task, the knee should perform the extension and flexion movements, respectively. Based on these assumptions, the knee exoskeleton provided a closed-loop torque control to extend and flex the knee joint, assisting the user during the first and the second subphases of the SA task. For the stance phase of the LW and SD tasks, the muscles around the knee joint provide negative power to help the knee move and avoid excessive knee flexion, supporting the human body. For this reason, in [9], an open-loop damping control was adopted to provide resistance at the knee joint to support the body and absorb the shock. In [33], contrary to the above-mentioned studies [4,7,9,32], the high level did not present a gait-phase estimation algorithm. The mid-level was responsible for traducing EMG signals to knee joint torques using a proportional gain method (EMG-based torque method) proposed by [48]. These estimated knee joint torques acted as reference signals in the low level, being compared with the measured knee joint torques to adapt the knee orthosis assistance.

#### 3.3.5. Participants

The studies included at least 91 participants for developing and validating the LM decoding tools from the 18 included studies. The maximum number of participants was 18 [5], the minimum was 1 [12], and the mode was 3 [3,6,8,10,11,34]. Moreover, from the 91 participants, 90 presented a healthy condition state, while only 1 pathological participant (stroke patient) was considered [9].

## 4. Discussion

This literature review shows that several studies focused on developing LMs recognition and prediction tools for gait rehabilitation purposes driven by AOs and exoskeletons. Apart from the general overview of the actual status, topics related to the studied LMs, sensors, type of input data, analysis windows, classifiers, and their performance and the control of the wearable assistive devices will be discussed to answer four raised RQs. In the end, the limitations of the current technologies are summarized, and future directions are proposed to tackle the identified challenges in this area.

### 4.1. Which Are the Typical LMs and the Target Population Addressed?

This review shows that LW, SA, and SD are the typical LMs addressed [3,4,5,6,7,8,9,10,11,12,13,30,31,32,34,35] (N = 16), while the LW, SA, SD, RA, and RD tasks correspond to the second most addressed tasks [4,8,11,12,13,30,31,32,35] (N = 9). These results are not following the findings of [1] since the most representative decoded tasks were the LW, SA, SD, RA, and RD, whereas the LW, SA, and SD were the second most explored tasks. This phenomenon may be associated with the fact that most of the studies reported in [1] use prostheses, which may reveal a different tendency compared to AOs and exoskeletons. The high prevalence of decoding dynamic tasks is associated with using these robotic devices to increase the motor independence of injured subjects in their daily lives. However, only three studies decoded the LW, SA, SD, RA, and RD tasks together and transitions between them [8,30,32]. The walking initiation and termination movements were only explored in [6,34]. Moreover, no studies decoded the ST-RA, RA-ST, ST-RD, and RD-ST transitions. Consequently, no study decoded all commonly daily performed LMs, including the LW, SA, SD, RA, RD, ST, and SIT tasks and transitions between them under robotic assistance. These facts are in accordance with findings from previous studies [21,23,24], in which the identification of transitions between several tasks when using a wearable assistive device is discussed as one of the main limitations of the current LM decoding tools.

Twelve studies [3,5,6,7,8,11,12,13,32,33,34,35] provided information about the gait speed, and in eight studies [3,5,6,7,8,32,34,35], the participants walked at self-selected speeds (not controlling it), and in four studies [11,12,13,33], the speed was fixed and controlled. These results follow the ones reported in [1] since there were more studies in which the gait speed was self-selected. Furthermore, according to [49], the average self-selected speed of neurologically impaired patients is about 0.46 m/s (1.66 km/h). Considering this information and the used gait speeds, only the study [33] seems to address the typical walking speeds of these patients. There is evidence that waking speed affects the lower limb biomechanics [50,51]. Consequently, the LM decoding tools’ performance may be jeopardized if walking speeds different from those used during the algorithms’ training process (commonly involving healthy subjects walking at higher self-selected speeds than patients) are employed [52,53,54]. Thus, the applicability of the available solutions trained with healthy gait patterns to pathological individuals may be compromised. Additionally, only one study from the reviewed studies involved a stroke patient for tool development, which may be insufficient to validate the application of the developed algorithms in this pathological population. This assumption is in accordance with [7,14], stating that the application of LM decoding algorithms developed for healthy participants in pathological subjects may not properly work since the biomechanics of pathological patients are different from healthy subjects. The recommendations suggest including the target population during the algorithms’ development [14].

### 4.2. Which Type of Wearable Sensors and Features Are Commonly Used for LM Recognition and Prediction?

Kinematic sensors are the most used for LM decoding. Among potentiometers, encoders, AHRS, and IMU sensors, these last ones correspond to the most used type of sensor, as depicted in Figure 3a–d. Based on the considered literature, physiological sensors are the sensor type less used for LM decoding. These results support the ones reported in [1,14,15]. Although EMG signals may allow recognizing the user’s motion intention faster due to their anticipatory ability (about 100 ms before the muscle contraction [14]), EMG-based approaches have been left behind since EMG sensing is prone to fade during long-term use as a result of (i) movements between the skin and the electrodes; (ii) temperature variations; and (iii) sweating [7,12,13,32,33,55,56]. These phenomena can cause an incorrect identification of the user’s LM. Moreover, according to [14,15], the use of EMG signals to decode LMs of pathological users (such as stroke patients) is prone to provide low accuracies due to the muscular activities of pathological users, which may vary across time and during the execution of the LMs as a result of fatigue. For this reason, the target population should be included during the algorithms’ training [14].

Considering the findings of [1,14], the classification accuracy for LM decoding algorithms may increase when data from kinematic, kinetic, and/or EMG systems are fused. The review [14] found that fusing data from EMG and IMU sensors is profitable since the effect of the sensor position in the user’s limb (which affects the EMG signals recorded) may be compensated by the position information provided by IMU sensors. Additionally, as reported in [1], the accuracy of classification algorithms fed by data from IMU and kinetic sensors is higher than those fed exclusively using IMUs. On the other hand, based on the reviewed studies, classifiers that only used IMU sensors [5,9,10,11,12,32,34] achieve, in general, similar classification accuracy to the ones that fused data from IMU sensors and kinetic sensors (load cells [13,35], FSRs [4,31,57], and GRF sensors [30] embedded into the assistive device). In some cases [5,10,32], the exclusive use of IMU sensors seems to provide higher average accuracies when compared to the combination of IMUs and other sensors (Figure 3b,c), which does not support the findings of [1]. The results of Figure 3b,c,d show that it is possible to recognize the most daily performed LMs (ST, LW, SA, SD, RA, and RD tasks) and the transitions between them (LW-SA, LW-SD, LW-RA, LW-RD, SA-LW, SD-LW, RA-LW, and RD-LW) only using IMUs [5,9,10,11,12,32].

Although combining IMUs with other sensors does not seem to provide higher accuracy, this combination seems to provide meaningful advances in terms of decoding time. In [8], lower recognition delays were obtained by fusing kinematic (AHRS sensors placed on the shank and foot segments) and kinetic (pressure insoles) sensors, adding the ability to predict the LW task when preceded by the SA task and vice-versa. Moreover, the fusion between pressure insoles with encoders typically embedded in the wearable assistive device to measure the joint angles appears to contribute to the decoding of other tasks and transitions, namely: SIT, ST, SIT-ST, ST-SIT, LW-ST, ST-LW, SA-ST, SD-ST, ST-SA, and ST-SD. In addition to being an essential contribution to recognizing the referred tasks, the fusion of pressure insoles with encoders in [6] allowed the ability to predict the SIT task preceded by ST task and vice-versa before their occurrence.

Further, raw temporal data directly measured by the sensors were the most common input data for LM decoding tools. Eight studies [2,3,4,5,6,13,30,31] used raw temporal data extracted from sensors, namely: (i) hip, knee, and ankle joint angles depending on the assisted joint [2,3,6,32]; (ii) inclination angles of the thigh, shank, and foot [4,13,31]; and (iii) vertical position and velocity of the striking foot [3,13,31]. This may be related to the time consumption associated with feature determination. The remaining studies used an analysis window to compute time- [5,7,8,9,10,11,12,34,35] (N = 9) and frequency-domain [8] (N = 1) features. The dominance of time-domain over frequency-domain features was also reported in [1]. It may be associated with the fact that the time-domain features are easier and faster to be computed. The most used time-domain features were (i) the hip joint angles and CoP at the heel-strike event [7]; (ii) maximum and minimum thigh, shank, and knee angles [5]; and (iii) maximum, minimum, mean, standard deviation, and root mean square of the thigh and shank inclination angles and triaxial angular velocities and accelerations [9,10,11,12], while the reported frequency-domain features were the (i) GRF during the swing phase [8] and (ii) thigh and foot inclination angles [8].

According to the findings reported in [1], it is preferable to use windows with a length varying between 100 and 250 ms when mechanical sensors (such as kinematic or kinetic) are used. Based on the seven studies that presented information regarding the analysis windows, various combinations were used: (i) window length of 100 ms with an increment of 10 ms [34,35]; (ii) window length of 100 ms with an increment of 50 ms [32]; (iii) window length of 150 ms with an increment of 10 ms [10]; (iv) window length of 200 ms with an increment of 10 ms [8]; and (v) window length of 250 ms and an increment of 10 ms [11,12]. These findings align with those reported in [1] since the analysis windows match the range from 100 to 250 ms.

### 4.3. Which Set of Algorithms Should Be Employed to Recognize/Predict Different LMs Attending to Accuracy and Time-Effectiveness?

Generally, the algorithm’s performance for LM decoding is evaluated based on two metrics: (i) the accuracy of the classification process and (ii) the recognition delay, which represents the period between the instant in which the locomotion mode starts and the instant in which that locomotion mode is recognized. Typically, this recognition delay is evaluated as a percentage of a gait cycle, and it should be as low as possible [23]. Ideally, the recognition delay would need to be negative, indicating that the motion task was predicted, i.e., recognized before its occurrence.

The choice between each classifier depends on the purpose of each study. Attending to the accuracy values presented in Table 2, the FSM, MFNN, and CNN may be the most appropriate classifiers to decode static LMs, namely ST and SIT tasks, since until now, they were the only algorithms applied for this purpose [6,7,9,11,12,34,35]. In addition, if the goal is to distinguish static tasks and transitions between static tasks (ST, SIT, SIT-ST, and ST-SIT), then it becomes more feasible to choose the FSM [6,7,9]. On the other hand, if the goal is to distinguish continuous dynamic tasks, MFNN, FSM, and DT classifiers appear to provide a higher capability (Figure 3a) [4,13,30,31]. Additionally, to decode continuous dynamic tasks and static tasks together (ST, LW, SA, SD, RA, and RD), FSM or MFNN may be preferable (Figure 3b) [3,11,12]. Otherwise, if the goal is to distinguish dynamic tasks and transitions between dynamic tasks, the SVM, CNN, kNN, stacked autoencoder-based deep neural network, or FSM may be employed [2,8,9,10,30,32] (Figure 3c). At last, to decode static and continuous tasks and transitions between them, the FSM combined with the SVM classifier may be the best option to take (Figure 3d) [9]. This different choice between classifiers considering the LM to decode depends not only on the classifier performance but also on the input information used. For example, CNN has the capacity to distinguish dynamic tasks and transitions between them with high accuracy only when fed by IMUs data. If CNN is fed by IMU, encoder, and GRF forces information together, the classification accuracy seems to drop, as depicted in Figure 3c [30,32]. For this reason, the data used to feed the algorithm present an important role in the algorithm’s performance.

Considering the recognition delay values stated in Table 2, most of the LMs were recognized between 3.96% and 100% of the gait cycle in the new LM, which means that the new LM is identified between 3.96% (after its beginning) and 100% (the end of the first gait cycle) of the new LM. This phenomenon may cause perturbations during the assistance due to the existent delay in identifying the new LM. Thus, timely assistance according to the user’s needs may not be achieved. Nonetheless, there were four studies able to predict some tasks before their occurrence, in which the SVM classifier stands out [6,8,9,10], achieving a prediction from 5.4 to 78.5 ms before the new LM [9,10]. Moreover, different decoding times were achieved when the leading leg was with or without the wearable assistive device. In [10], the SA and SD tasks preceded by the LW task were predicted 5.4 ± 74.6 ms and 78.5 ± 25.0 ms before their occurrence when the leading leg was the leg without the wearable assistive device. In the same study [10], the SD task was predicted by 1.8 ± 39.2 ms in advance, while the LW task was predicted 46.7 ± 46.6 ms before its occurrence. These results support that the leading leg may affect the temporal performance of the classifier.

### 4.4. How to Adapt the Exoskeleton/Orthosis Assistance According to the Decoded User’s LM

Considering the findings of this review, five studies adapted the assistance of the wearable assistive device attending to the user’s decoded LM [4,7,9,32,33].

Based on the collected information, the provision of assistance mostly depends on both ongoing LM and gait events, namely stance and swing phases or specific gait events, such as heel-strike, heel-off, and toe-off. Two designs can be employed to adapt the wearable assistive device assistance according to the decoded LM, namely a two-level [4,7,34] and a three-level [9,32,33] hierarchical control. While in [4], a constant torque of 10 N.m was provided by an ankle orthosis when the SD task was recognized, study [7] adapted the joint torque trajectories from [42,43,44,45] according to the recognized tasks (SIT, ST, LW, SA, and SD tasks and transitions between them). Apart from the difference in the torque trajectories, the highest distinction between the control of [4] and [7] is that, while in [4], there was a gait-phase estimation algorithm running before the LM recognition tool to provide features for LM decoding, in [7], the gait-phase estimation algorithm was running after the LM recognition to set the correct time to adapt the assisted torque. Thus, the development of LM decoding tools dependent on preliminary gait algorithms may affect the recognition delay and, consequently, the time compliance of the wearable assistive device’s control. This phenomenon may be the reason for the one-step delay (100% of the gait cycle) exhibited in [4], which is higher than the recognition delay presented in [7] (from 15.2 to 63.8% of the gait cycle). Moreover, despite being a promising implementation, the feasibility of the assistance provided in [7] may be compromised because adopting trajectories from the literature should be done carefully since these trajectories depend on walking speed and the user’s anthropometry [50]. According to [4], the ideality would be to have a library with different trajectory profiles related to various motion tasks, walking speeds, and user’s anthropometry embedded in the control scheme. This would be a valuable contribution to selecting the required trajectory to assist the user according to their motion intentions.

In [9,32,33], a three-level hierarchical control architecture was followed. While in [32], the high level presented a gait-phase estimation algorithm followed by the LM recognition tool, in [9], the LM recognition tool was not dependent on the gait-phase detection algorithm. As reported in the studies [4,7], the non-dependence of previous gait analysis tools may explain the ability of the study [9] to predict some LMs (−78 ms to 38 ms), whereas no prediction was reported in [10] (3.96 to 24 of the gait cycle). Regarding the mid-level, different choices were made in each study, with the following being used: (i) a POILC method for assisting the LW, SA, and SD tasks [32]; (ii) a hybrid torque method for assisting the ST, LW, SA, and SD tasks [9]; and (iii) an EMG-based torque method for assisting the LW task [33]. It is not possible to suggest a control scheme to assist a specific motion since there is no benchmarking analysis of controllers’ performance when considering different LMs. However, according to [52], a hybrid control approach should be adopted to assist the user or reduce the constraints associated with the actuator’s frictions only when needed.

Providing efficient assistance according to the decoded user’s LM implies an accurate and timely identification of the user’s intentions. This is of utmost importance since the higher the anticipation time in identifying the LM, the more time remains to switch the control to assist the users according to their needs timely. This means that, ideally, the decoded LM should be recognized before its occurrence, i.e., predicted, to adapt the assistance according to the LM identified. Considering the five studies that adapt the assistance according to the decoded LM, only one study [9] enables the prediction of the LW-SA, SA-LW, LW-SD, and SD-LW transitions before their occurrence. This may be related to the non-dependence of the LM recognition algorithm on gait event detection algorithms.

Current directions recommend the use of co-adaptive control assistance, named as the “Assist-as-Needed” (AAN) approach, in which the patients are encouraged to participate in the rehabilitation tasks, and the wearable assistive device only assists when and as much as required, helping the users to accomplish a specific motion [58,59,60]. However, based on the collected studies, even those that adapt the assistance according to the user’s motion intention [4,7,9,32,33], no study followed an AAN approach. Thus, the adoption of the current LM-driven control strategies during the rehabilitation of neurologically impaired patients may be compromised.

### 4.5. Review Limitations

This study aims to present a systematic review regarding the state of LM decoding tools when using wearable assistive devices, such as lower-limb AOs and exoskeletons. Thus, studies in which the participants were not wearing any wearable assistive device (exoskeletons or AOs) or using prosthetic systems were excluded. Additionally, the literature search was performed using *Scopus* and *Web of Science* databases, including studies published after 2010 and written in English. Therefore, there are possibly other studies before 2010 belonging to other databases and written in other languages that were not included.

Further, considering the studies that predicted some LMs, it was not possible to present the accuracy value concerning the recognized and predicted LM separately since the studies only provided the final mean value of the accuracy for all decoded LMs, covering the recognized and the predicted ones [6,8,9,10].

### 4.6. Suggestions for Future Research

This review included 18 papers for LM recognition and prediction when using wearable assistive devices. Despite revealing meaningful improvements in this field, there is room for new developments and improvements in an attempt to develop smarter exoskeletons and AOs for personalized assistance. Based on the findings reported in this review, we report some highlights and suggestions to design time-effective LM decoding tools based on meaningful data for real-time integration into robotic devices to adapt the assistance according to the user’s intent.

Firstly, studies should develop and validate the decoding tools considering the conditions used to deploy the developed tools with the end-users and in real environments. However, most of the studies have tested their solutions in healthy subjects, exhibiting insufficient clinical evidence. Since only one study [9] involved a pathological participant in developing the LM decoding tool, it is not yet possible to guarantee the versatility and feasibility of these tools when considering a pathological population. According to [7,14], an algorithm for LM decoding developed for healthy gait patterns could be prone to failure when applied to pathological gait patterns because the biomechanics of pathological patients (such as stroke patients) are modified due to their health status. This phenomenon reinforces that the available solutions for LM decoding are not ready to support the daily living of pathological patients when using wearable assistive devices.

Secondly, the available tools, even those that adapt the assistance of the wearable assistive device considering the decoded LM, did not implement an AAN scheme in which the wearable assistive device increases or decreases its contribution based on the user’s motor performance. The absence of this seamless human–machine interaction may not encourage the participation of neurologically impaired patients during rehabilitation tasks. Thus, personalized assistance may not have been achieved yet. Therefore, it is necessary to develop more studies with patients with lower-limb impairments walking at their self-selected walking speeds, involving them in AAN control schemes.

Thirdly, identifying transitions between several tasks when using a wearable assistive device represents one of the main limitations of the current tools. Therefore, efforts should address the lack of information in decoding the walking initiation and termination and the ST-RA, RA-ST, ST-RD, and RD-ST transitions. Fourthly, until now, only the SA, LW, ST, and SIT transitions were predicted before their occurrence when using wearable assistive devices. Moreover, several tasks are recognized after their occurrence. This may introduce delays in the LM decoding scheme, which may harm the provided assistance in relation to real-time constraints. Thus, required innovations should tackle the emergent need for developing user-independent and generic LM decoding tools that enable real-time motion intention monitoring in free-living scenarios with high accuracy and a predictive character.

## Figures and Tables

**Figure 1 sensors-22-07109-f001:**
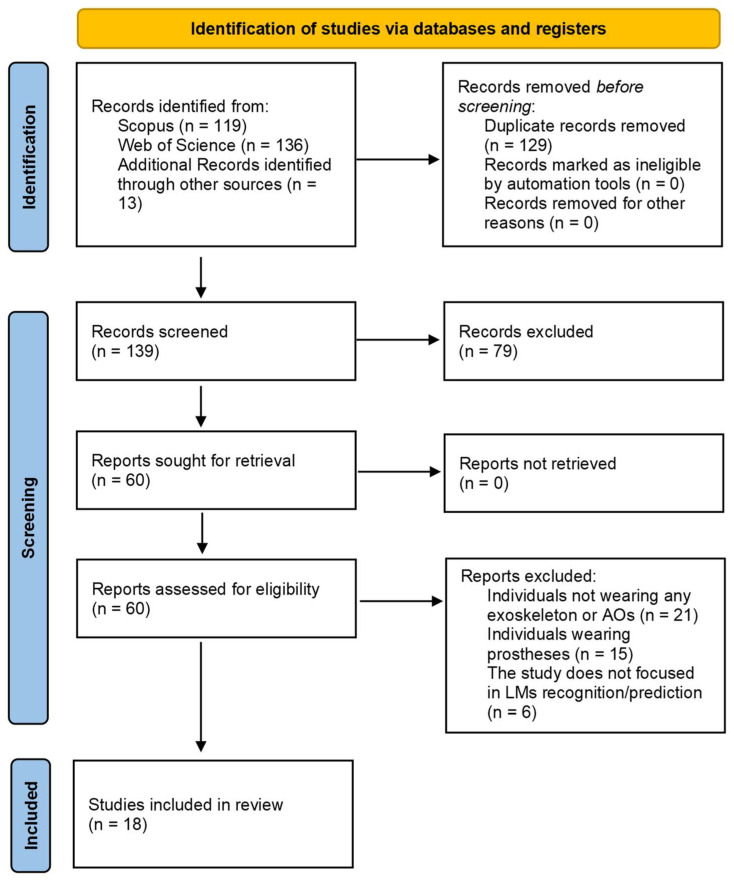
PRISMA flow chart for LM recognition and prediction.

**Figure 2 sensors-22-07109-f002:**
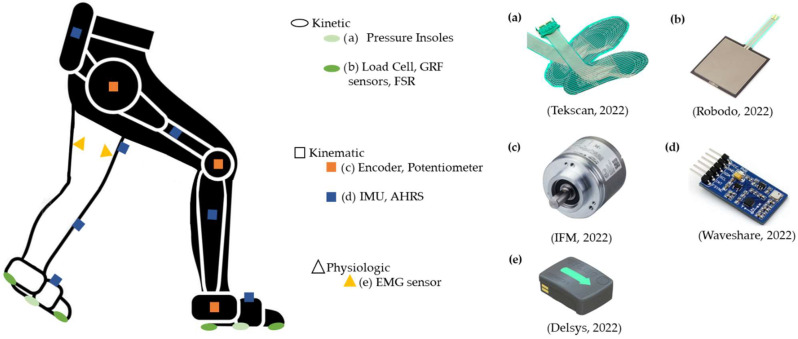
Representation of the kinetic [36,37], kinematic [38,39], and physiologic [40] sensors found in the literature. The assisted and non-assisted legs are represented in black and white colors, respectively.

**Figure 3 sensors-22-07109-f003:**
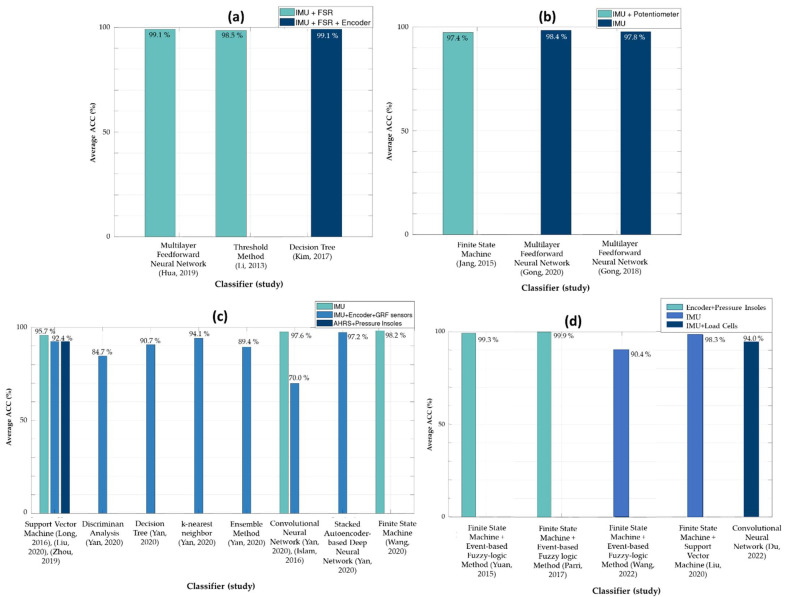
Average accuracies reported by: (**a**) studies [4,13,31] able to decode continuous dynamic LMs; (**b**) studies [3,11,12] able to decode continuous dynamic and static LMs; (**c**) studies [5,8,9,10,29,31] able to decode continuous and discrete dynamic LMs; and (**d**) studies [6,7,9,34,35] able to decode static, continuous, and discrete LMs. The standard deviation was not presented for reasons of simplicity.

**Figure 4 sensors-22-07109-f004:**
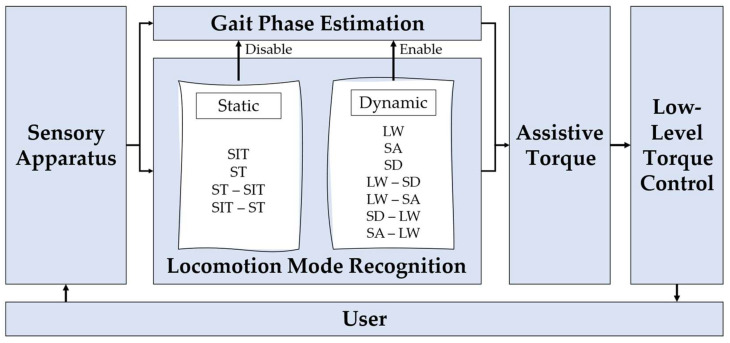
Control scheme adopted in [7]. For continuous dynamic tasks, the gait phase estimation algorithm was performed after the LM decoding and before the provision of assistive torque. Adapted from [7].

**Table 1 sensors-22-07109-t001:** Quality assessment of the included studies.

Study	Criterion												Score (%)
	C1	C2	C3	C4	C5	C6	C7	C8	C9	C10	C11	C12	
[7]	2	2	1	1	2	2	1	2	2	2	2	2	87.5
[13]	2	2	2	2	2	2	0	2	2	2	2	2	91.7
[6]	2	2	1	1	2	2	0	2	2	2	2	2	83.3
[10]	1	2	1	1	2	2	2	2	2	2	2	2	87.5
[30]	1	2	0	1	2	1	0	2	2	0	2	2	62.5
[8]	1	2	2	1	2	2	2	2	2	2	2	2	91.7
[31]	1	2	1	1	2	2	0	2	1	2	1	2	70.8
[3]	2	2	1	1	2	2	0	2	1	2	2	2	79.2
[32]	2	2	1	1	2	2	2	2	2	2	2	2	91.7
[11]	2	2	1	2	2	2	2	2	1	2	1	2	87.5
[12]	2	2	2	2	2	2	2	2	1	2	2	2	95.8
[4]	2	2	1	1	2	2	0	2	2	2	2	2	83.3
[9]	2	2	1	1	2	2	1	2	2	2	2	2	87.5
[33]	1	2	2	2	2	2	0	2	2	2	1	2	83.3
[5]	2	2	2	1	2	2	0	2	2	2	2	2	87.5
[2]	2	2	1	1	2	2	0	2	2	2	1	0	70.8
[34]	2	2	2	2	2	2	2	2	2	1	2	2	95.8
[35]	2	2	1	2	2	2	2	2	0	2	2	2	87.5
								**Mean Score ± standard deviation**					**84.7 ± 8.7**

**Table 2 sensors-22-07109-t002:** LM decoding algorithms available in the literature.

Study	R/P ^1^	Locomotion Tasks	Speed	Sensors (Location)	Features	Windows	Classifier	Performance (ACC ^2^, Delay)	Control Type	Participants (Status)
Parri et al. [7]	R	Static Tasks: SIT ^3^, ST; Dynamic Tasks: Continuous (LW, SA, and SD) and Transitions (LW→ST, ST→LW, SIT→ST, ST→SIT, LW→SA, LW→SD, SA→LW, SD→LW)	Slow, natural, and fast	-Encoder (hip exoskeleton) -Pressure insoles (feet)	-Hip joint angles and Center of Pressure (CoP) at specific gait events	200 ms	-Static and Discrete Tasks: Finite State Machine (FSM);	-ACC > 97.4%; -Dynamic Motion: Delay about one step; -Transitions: Delay between 15.2% and 63.8%	-Zero-torque mode; -Assistive mode without considering the motion intention	6 (healthy)
Kim et al. [13]	R	Dynamic Tasks: Continuous (LW, SA, SD, RA, and RD)	Fixed speed (4 km/h)	-Encoder (hip and knee exoskeleton); -5 IMU (exoskeleton back, thigh, and foot); -Load cells (exoskeleton insole)	-Vertical foot position -Thigh, shank, and foot inclination	NI ^4^	Decision Tree (DT)	-Average ACC = 99.1%; -Delay between 0.0% and 5.13%	Zero-torque mode	8 (healthy)
Yuan et al. [6]	Both	Static Tasks: SIT, ST; Dynamic Tasks: Continuous (LW, SA, SD) and Transitions (SIT→ST, ST→SIT, ST→LW, LW→ST, ST→SA, SA→ST, ST→SD, SD→ST, LW→SA, SA→LW, LW→SD, SD→LW)	Natural	-Encoder (hip exoskeleton); -Pressure insoles (feet)	-Hip joint angles and Center of Pressure (CoP) at specific gait events	NI	-Static Tasks and Transitions: FSM; -Continuous Tasks: Event-based fuzzy-logic method;	-ACC > 90.1%; -Delay between −30.9% and 100%	Zero-torque mode	3 (healthy)
Zhou et al. [10]	Both	Dynamic Tasks: Continuous (LW, SA, SD) and Transitions (LW→SA, SA→LW, LW→SD, SD→LW)	NI	2 IMU (exoskeleton thigh and shank)	Maximum (MAX), minimum (MIN), mean, standard deviation, and root mean square (RMS) of the thigh inclination angles, angular velocities, and angular accelerations	150 ms with an increment of 10 ms	Support Vector Machine (SVM)	-ACC between 93.0% and 96.2%; -Delay between −40.0 ms and 185 ms;	Zero-torque mode	3 (healthy)
Hua et al. [30]	R	Dynamic Tasks: Continuous (LW, SA, SD, RA, RD) and Transitions (LW→SA, SA→LW, LW→SD, SD→LW, LW→RA, RA→LW, LW→RD, RD→LW)	NI	-Encoder (exoskeleton); -1 IMU (exoskeleton back); -Ground Reaction Force (GRF) sensors (exoskeleton);	NI	NI	-DT; -Discriminant Analysis (DA); -SVM; -k-Nearest neighbor (KNN); -Ensemble Method (EM); -Convolutional Neural Network (CNN); -Stacked Autoencoder-based Deep Neural Network	-ACC = 99.7%; -Delay between 11.8% and 17.4%;	NI	NI
Long et al. [8]	Both	Dynamic Tasks: Continuous (LW, SA, SD, RA, RD) and Transitions (LW→SA, SA→LW, LW→SD, SD→LW, LW→RA, RA→LW, LW→RD, RD→LW)	Natural	-2 Attitude and Heading Reference System (AHRS) sensors (shank and foot); -6 GRF sensors (pressure insoles/feet);	Wavelet coefficients from (i) GRF during the swing phase; and (ii) thigh and foot inclination angles	200 ms with an increment of 10 ms	SVM	-ACC between 97.3% and 99.5%; -Delay between −10.4% and 48%	Zero-torque mode	3 (healthy)
Islam et al. [31]	R	Dynamic Tasks: Continuous (LW, SA, SD, RA, RD)	NI	-1 IMU (orthosis foot); -Force Sensor Resistor (FSR) (orthosis insole);	-Vertical foot position -Foot orientation -FSR-based foot contact information	NI	Multilayer Feedforward Neural Network (MFNN)	-ACC > 98.3%; -Delay between 16% and 28%	Zero-torque mode	5 (healthy)
Jang et al. [3]	R	Static Tasks: ST; Dynamic Tasks: Continuous (LW, SA, SD)	Natural	-Potentiometers (hip exoskeleton); -1 IMU (exoskeleton back);	-Hip joint angles -Vertical acceleration-based foot contact	NI	FSM	-ACC between 95% and 99%; -Delay of one-step delay	Zero-torque mode	3 (healthy)
Zhu et al. [32]	R	Dynamic Tasks: Continuous (LW, SA, SD, RA, RD) and Transitions (LW→SA, SA→LW, LW→SD, SD→LW, LW→RA, RA→LW, LW→RD, RD→LW)	Natural	4 IMU (thigh and shank)	Hip and knee joint angle, angular velocity, and angular acceleration	100 ms with an increment of 50 ms	CNN	-ACC between 96.6 and 99.0%; -Delay between 3.96% and 24.0%	Assistive mode considering the motion intention	7 (healthy)
Gong et al. [12]	R	Static Tasks: ST; Dynamic Tasks: Continuous (LW, SA, SD, RA, RD)	Fixed speed (2.7 km/h)	2 IMU (thigh)	MAX, MIN, mean, standard deviation, and RMS of the thigh inclination angles, angular velocities, and angular accelerations	250 ms with an increment of 10 ms	MFNN	-Average ACC = 97.8% -Delay between 50 and 300 ms	Zero-torque mode	1 (healthy)
Gong et al. [11]	R	Static Tasks: ST; Dynamic Tasks: Continuous (LW, SA, SD, RA, RD)	Fixed speed (2.7 km/h)	2 IMU (thigh)	MAX, MIN, mean, standard deviation, and RMS of the thigh inclination angles, angular velocities, and angular accelerations	250 ms with an increment of 10 ms	MFNN	-Zero-torque mode: Average ACC = 98.4%; -Assistive mode: ACC between 97.6% and 98.4%;	-Zero-torque mode; -Assistive mode without considering the motion intention;	3 (healthy)
Li et al. [4]	R	Dynamic Tasks: Continuous (LW, SA, SD, RA, RD)	NI	-1 IMU (orthosis foot) -FSRs (orthosis insole)	-Orthosis orientation -Orthosis position	NI	FSM	-ACC between 97.2% and 99.5%; -Delay of one-step delay	Assistive mode considering the motion intention	5 (healthy)
Liu et al. [9]	Both	Static Tasks: ST; Dynamic Tasks: Continuous (LW, SA, SD, RA, RD) and Transitions (LW→SA, SA→LW, LW→SD, SD→LW, LW→RA, RA→LW, LW→RD, RD→LW)	NI	2 IMU (exoskeleton thigh and shank)	MAX, MIN, mean, standard deviation, and RMS of the thigh and shank inclination angles, angular velocities, and angular accelerations	15 samples	-Static Tasks and Transitions: FSM; -Continuous Tasks: SVM;	-Healthy participants: average ACC between 97.6% and 98.3% and delay between −78.5 ms and 38.7 ms; -Stroke participant: average ACC = 97.4%;	Assistive mode considering the motion intention	-5 (healthy); -1 (stroke);
Fernandes et al. [33]	R	Dynamic Task: Continuous (LW)	Fixed speed (1 km/h and 1.5 km/h)	Electromyography (EMG) (Vastus Lateralis, Vastus Medialis, Semitendinosus, and Semimembranosus)	EMG data from Vastus Lateralis, Vastus Medialis, Semitendinosus, and Semimembranosus	NI	Proportional Gain Method	-NRMSE = 12%; -Delay = 22 ms;	Assistive mode considering the motion intention	2 (healthy)
Wang et al. [5]	R	Dynamic Tasks: Continuous (LW, SA, SD) and Transitions (LW→SA, SA→LW, LW→SD, SD→LW)	Natural	2 IMU (thigh and shank)	-MAX and MIN thigh and shank angles; -MAX and MIN knee angles;	NI	FSM	-ACC between 98.1% and 98.3%; -Delay between 41.1% and 58.2%;	Zero-torque mode	18 (healthy)
Kimura et al. [2]	R	Dynamic Tasks: Transitions (SIT→ST, ST→SIT)	NI	Potentiometer (hip and knee exoskeleton)	-Hip and knee joint angle -Upper body pitch angle	NI	SVM	-F-Measure between 0.882 and 0.997	Zero-torque mode	6 (healthy)
Du et al. [34]	R	Static Tasks: ST; Dynamic Tasks: Continuous (LW, SA, SD) and Transitions (ST→LW, LW→ST)	Natural	2 IMU (thigh)	Pitch and roll angles	100 ms with an increment of 10 ms	-Static Tasks and Transitions: FSM; -Continuous Tasks: Event-based fuzzy-logic method;	-ACC of 91.9% between static tasks and ACC higher than 89.0% between dynamic tasks; -Delay = 554.4 ms	Zero-torque and Assistive mode considering motion intention	3 (healthy)
Wang et al. [35]	R	Static Tasks: ST and SIT; Dynamic Tasks: Continuous (LW, SA, SD, RA, RD) and Transitions between Static Tasks	Natural	-6 IMU (thigh, shank, and shoes) -4 Load cells (insole)	NI	100 ms with an increment of 10 ms	CNN	-ACC = 94.0%; -Delay between 18.1 and 53.3 ms	Zero-torque mode	9 (healthy)

^1^ R and P mean recognition and prediction, respectively, ^2^ ACC means accuracy, ^3^ Sitting (SIT), Standing (ST), Level-ground Walking (LW), Stair Ascending (SA) and Descending (SD), Ramp Ascending (RA) and Descending (RD), ^4^ NI means Not Indicated. It was used when the information was not provided in the studies.

## Data Availability

Not applicable.

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
