# Peer review of "A Review on Locomotion Mode Recognition and Prediction When Using Active Orthoses and Exoskeletons"

_sensors, 2022, doi:10.3390/s22197109_

Round 1

Reviewer 1 Report

The review article is well prepared, well prepared, and well written. The number of references is pretty few for the typical review; however, it is well described that the review criteria exclude several references based on reasonable filters.

Overall, the review article is impressive and can be a good reference for new scientists and experts in the area of Locomotion Mode Recognition and Prediction.

The reviewer highly recommends the article for acceptance for publication in sensors.

Author Response

Dear Reviewer,

Thank you for reviewing our manuscript submitted to Sensors.
We are glad to see that the reviewer recommends the manuscript for publication. Thank you for your feedback.
We would like to inform you that a new search was done, and the manuscript was updated with two more papers for review analysis. The added information is highlighted in the manuscript. Now, the review is updated to August 2022.

Best regards,
The authors

Reviewer 2 Report

After studying the article, I can only positively evaluate the topicality of the article. The article has a poorly designed article selection methodology. A selection of only 16 studies is not enough for this type of publication. I recommend that the article be rejected.

Author Response

Dear Reviewer,

Thank you for reviewing our manuscript submitted to Sensors.
We have considered the reviewer's comments. The followed review methodology is according to the one conducted by previous works [1], [2], including reviews published in Sensors [3]. Further, we believe that the eligibility criteria are appropriate and compliant with the goals of this review.
The focus was given to wearable technologies (both robotic assistive devices and sensors) due to the following aspects: (i) it meets the clinical recommendation for functional gait training in locomotor activities related to the patients’ daily living activities, so they get trained for motor independence; (ii) it allows to conduct more versatile and realistic locomotion modes in both indoor and outdoor environments and, consequently, to create decoding algorithms more robust for application in real scenarios; (iii) wearability may imply a more challenging locomotion mode classification than the one achieved with non-wearable devices; thus a concurrent analysis could be unfair; (iv) it focuses on active orthoses and exoskeletons rather than prostheses since the locomotion mode recognition performed when using orthotic/exoskeleton systems is different from the one performed when using prostheses due to the distinct positions that the sensors can assume in these situations. We consider that the conclusions of this review are relevant for stakeholders on wearable assistive devices.
We would like to inform you that we conducted a new search following the same methodology, adding two more papers to the review analysis. The added information is highlighted in the manuscript. Now, the review is updated to August 2022.

Best regards,
The authors

References

[1]      T. Yan, M. Cempini, C. M. Oddo, and N. Vitiello, “Review of assistive strategies in powered lower-limb orthoses and exoskeletons,” Rob. Auton. Syst., vol. 64, pp. 120–136, Feb. 2015, doi: 10.1016/j.robot.2014.09.032.

[2]      D. Novak and R. Riener, “A survey of sensor fusion methods in wearable robotics,” Rob. Auton. Syst., vol. 73, pp. 155–170, Nov. 2015, doi: 10.1016/j.robot.2014.08.012.

[3]      F. Labarrière et al., “Machine Learning Approaches for Activity Recognition and / or Activity Prediction in Locomotion Assistive Devices — A Systematic Review,” Sensors, pp. 1–30, 2020.

Reviewer 3 Report

In this brief review titled: A Review on Locomotion Mode Recognition and Prediction When Using Active Orthoses and Exoskeletons, the authors performed a systematic review by electronic search in Scopus and Web of Science databases, including published studies from 1 January 2010 to 31 June 2021. Sixteen studies were included and scored with 84.1±8.4% quality. Decoding focused on level-ground walking along with ascent/descent stairs tasks performed by healthy. Time-domain raw data from Inertial Measurement Unit sensors were the most used data. Different classifiers were employed consid-ering the LMs to decode (accuracy above 90% for all tasks). Five studies have adapted the assistance of AOs/exoskeletons attending to the decoded LM, in which only 1 study predicts the new LM before its occurrence. Future research is encouraged to develop decoding tools considering data from people with lower-limb impairments walking at self-selected speeds while performing daily LMs with AOs/exoskeletons. I believe this work will be of significant interest for not only this filed but wide audiences. 

However, there are still some minor concerns before this manuscript should be published in Sensors.

1) There are many grammatical errors and typos in the text. Please polish the language again.

2) Some figures are recommended to merge together instead of listed separately. 

3) More digital images (especially sensing or sensors instruments) adapted for existing literatures are recommend to put in the review instead of data sheets.   

4) Missing comparison with existing studies, especially sensing and sensors. More highly impacted and up to date research works should be cited and discussed well.

5) Some already published review in this Journal are recommend to read carefully for reference to avoid style problems, authors should read more reviews before writing. 

6) Some up to date related literatures should be discussed. Lab on a Chip, 2019, 19, 3602-3608.

Thanks.

Author Response

Dear Reviewer,

Thank you for reviewing our manuscript submitted to Sensors.
We have copied all your comments below and added our reactions to each suggestion. We have described how we incorporated the suggestions in the revised manuscript, highlighting in the manuscript all performed changes.
We hope that we have addressed your comments satisfactorily and that this revised manuscript can be accepted for publication.

Best regards,
The authors

Round 2

Reviewer 2 Report

The authors have edited the manuscript and it may be published in this form.